# Prediction of 72-hour mortality in patients with extremely high serum C-reactive protein levels using a novel weighted average of risk scores

**Kai Saito**[1], **Hitoshi Sugawara**[2]*, **Kiyoshi Ichihara**[3], **Tamami Watanabe**[2], **Akira Ishii**[2], **Takahiko Fukuchi**[2]

**1** Nara Medical University, Kashihara, Nara, Japan, **2** Division of General Medicine, Department of Comprehensive Medicine 1, Saitama Medical Center, Jichi Medical University, Saitama, Japan, **3** Faculty of Health Sciences, Yamaguchi University Graduate School of Medicine, Ube, Yamaguchi, Japan

* hsmdfacp@jichi.ac.jp

**Data Availability Statement:** All relevant data are within its Supporting Information files composed of Datasets S1, S2 and S3.

## Abstract

The risk factors associated with mortality in patients with extremely high serum C-reactive protein (CRP) levels are controversial. In this retrospective single-center cross-sectional study, the clinical and laboratory data of patients with CRP levels ≥40 mg/dL treated in Saitama Medical Center, Japan from 2004 to 2017 were retrieved from medical records. The primary outcome was defined as 72-hour mortality after the final CRP test. Forty-four mortal cases were identified from the 275 enrolled cases. Multivariate logistic regression analysis (MLRA) was performed to explore the parameters relevant for predicting mortality. As an alternative method of prediction, we devised a novel risk predictor, "weighted average of risk scores" (WARS). WARS features the following: (1) selection of candidate risk variables for 72-hour mortality by univariate analyses, (2) determination of C-statistics and cutoff value for each variable in predicting mortality, (3) 0–1 scoring of each risk variable at the cutoff value, and (4) calculation of WARS by weighted addition of the scores with weights assigned according to the C-statistic of each variable. MLRA revealed four risk variables associated with 72-hour mortality—age, albumin, inorganic phosphate, and cardiovascular disease—with a predictability of 0.829 in C-statistics. However, validation by repeated resampling of the 275 records showed that a set of predictive variables selected by MLRA fluctuated occasionally because of the presence of closely associated risk variables and missing data regarding some variables. WARS attained a comparable level of predictability (0.837) by combining the scores for 10 risk variables, including age, albumin, electrolytes, urea, lactate dehydrogenase, and fibrinogen. Several mutually related risk variables are relevant in predicting 72-hour mortality in patients with extremely high CRP levels. Compared to conventional MLRA, WARS exhibited a favorable performance with flexible coverage of many risk variables while allowing for missing data.

**Funding:** This work was supported by JSPS KAKENHI (Grant Number: 26460916), Pfizer Japan Academic Contribution 2017, 2018, and 2019, Astellas Research Support (Grant Numbers: RS2017A000020 and RS2018A000782), and Daiichi Sankyo Research Support Program 2018. HS received these grant awards. These funders had no role in study design, data collection and analysis, decision to publish, or preparation of the manuscript.

**Competing interests:** The authors have declared that no competing interests exist.

## Introduction

Clinical data are frequently collected in daily practice at medical institutions. Generally, laboratory data are used to generate alerts to improve clinical practice and ensure that the most appropriate care is provided to the patient. Reuse or secondary use of clinical laboratory data is an emerging field that is recognized as being essential for delivering high-quality healthcare and improving healthcare management [1, 2]. We examined the utility of laboratory data to determine how the structure of the medical care data marketplace can affect research priorities, gaps, and possibilities. Health information technology may have the potential to improve the collection and exchange of personal health records, allowing for utilization in electronic form [1, 2]. However, the reuse or secondary use of clinical data for the improvement of the overall quality of medical care remains limited.

A critical (panic) value is defined as a value that represents a pathophysiological state with extreme deviation from normal as it becomes life-threatening without prompt action [3]. Meanwhile, extreme outlier values are statistically expressed as below the 0.5 to 1.0 percentile value, or above the 99.0 to 99.5 percentiles [4]. These critical values have gained attention in the efficacy of identifying and communicating relevant information to treating physicians [5]. Conversely, physicians who encounter patients with extreme outlier laboratory values may be unsure whether such values are critical.

C-reactive protein (CRP) is a phylogenetically highly conserved plasma protein that functions as an acute inflammatory marker [6]. The median CRP concentration in healthy young adults is 0.08 mg/dL [7]. An acute stimulus very rapidly initiates *de novo* synthesis of CRP in hepatocytes, and plasma levels rise above 0.5 mg/dL within 4–8 hours, and peak at 48 hours. The half-life of plasma CRP is 19 hours, and a delay of 48 hours in reaching the maximum CRP value has been reported in critically ill patients [8]. CRP has versatile roles in both physiological and pathophysiological states [9], and has long been employed for clinical purposes as a biomarker for acute inflammation [7]. Recent studies have shown that CRP levels are related to the prognosis or activity of various diseases [10].

In hospital settings, physicians may encounter patients with CRP levels that significantly exceed the normal range [11]. Indeed, several case reports have reported CRP levels >40 mg/dL in adults with pyogenic liver abscess with complicated intestinal tuberculosis [12] and immune-hemolytic anemia [13]. Silvestre et al. observed no significant differences in CRP concentrations at ICU admission between survivors and non-survivors (25.3 ± 13.7 versus 28.2 ± 13.2 mg/dL). Furthermore, the ICU mortality rates of patients with sepsis during an ICU stay with CRP concentrations <10, 10–20, 20–30, 30–40, and >40 mg/dL were 20%, 34%, 30.8%, 42.3%, and 39.1%, respectively. This finding suggests that CRP is a poor marker of prognosis and that CRP levels >40 mg/dL are not associated with increased mortality of patients with sepsis during ICU stay [14]. In contrast, other studies have reported that CRP level >40 mg/dL indicate severe bacterial infection [15, 16], can be used as a predictor of renal scarring associated with a first urinary tract infection [17], and indicate acute pyelonephritis in the pediatric field [18]. Moreover, the cutoff value of CRP level >40 mg/dL was used to predict sepsis in emergency departments and demonstrated a sensitivity of 82.3% and specificity of 38.7% [19]. Furthermore, the use of a point-of-care CRP level >40 mg/dL significantly increases the predictive accuracy of treatment failure among patients with exacerbated mild to moderate chronic obstructive pulmonary disease [20, 21]. However, the prediction model of 72-hour mortality for patients with an extremely high outlier value of CRP level >40 mg/dL remains elusive.

In the current study, we aimed to elucidate the risk variables for predicting 72-hour mortality among patients with extremely high CRP levels under heterogeneous pathological

conditions. We primarily applied multivariate logistic regression analysis (MLRA) to explore the variables for predicting mortality. Furthermore, we devised a flexible method of predicting mortality by using a novel score, called "weighted average of risk scores" (WARS) as an alternative approach. The WARS is based on the summation of scores assigned to each risk variable in predicting the mortal outcome, and can incorporate all available risk variables, even with some missing data. Its performance and clinical utility, in contrast to the conventional MLRA, will be presented in predicting fatal outcomes among patients with extremely high CRP levels.

We anticipate that the development of a reliable numerical model for predicting mortality will contribute to improved initial management of severely ill patients in both primary and critical care settings.

## Materials and methods

The study protocol was designed according to the tenets of the Declaration of Helsinki [22] and was approved by the Institutional Clinical Research Ethics Review Board of Saitama Medical Center, Jichi Medical University, Saitama, Japan (Clinical #10–79 and #S20-025). The requirement for informed consent was waived due to the retrospective nature of the study.

### Study design and participant selection

This was a retrospective, single center, case-controlled cross-sectional study. We used a CRP data list that included 1,336,403 patients aged over 18 years who visited Saitama Medical Center between 2004 and 2017. The incidence of an extremely high outlier value of CRP level >40 mg/dL was 0.0302% overall, and 401 patients were selected. This rate meets the extreme outlier values that lie statistically outside the 0.5.99.5 percentile range [4]. We excluded 113 records that were metachronous duplicates from the same patient (only one highest value of CRP from each of these patients was considered), 10 patients with cardiopulmonary arrest at arrival, and 6 patients with unknown outcomes. After applying the exclusion criteria, a sample of 275 patients was selected for use as a training dataset (**S1 Dataset**) to build a model for predicting mortality risk.

Two additional patients' records were retrieved for validation of the model: one for confirmation (**S2 Dataset**; composed of 90 patients with CRP levels ≥40 mg/dL from a subsequent period between 2018 and 2020) and the other for assessing the specificity of risk variables included in the regression (**S3 Dataset**; composed of 818 patients with 20 ≤ CRP < 40 mg/dL and retrieved from 2019).

A flowchart of the selected cohort is shown in **Fig 1**.

The primary outcome was 72-hour mortality [23, 24] following the CRP test, regardless of whether the patient was hospitalized or in an outpatient setting. Cases were defined as patients with extremely high CRP levels who died in hospital during the first 72 hours after the test, while the controls were patients with extremely high CRP levels who survived.

The following risk factors were tested for their association with the 72-hour mortality outcome: age, sex, height, weight, body mass index (BMI), number of cigarettes smoked (Brinkman index), vital signs at the time of examination (e.g., systolic blood pressure, diastolic blood pressure, heart rate, respiratory rate, and body temperature), laboratory test values (e.g., white blood cell [WBC], red blood cell [RBC], hemoglobin [Hb], hematocrit [Ht], platelet [Plt], total protein, albumin [Alb], total bilirubin, direct bilirubin, aspartate transaminase [AST], alanine transaminase [ALT], γ-glutamyl transpeptidase [γ-GTP], lactate dehydrogenase [LDH], alkaline phosphatase [ALP], creatine kinase [CK], amylase, C-reactive protein [CRP], sodium [Na], potassium [K], chloride [Cl], calcium [Ca], inorganic

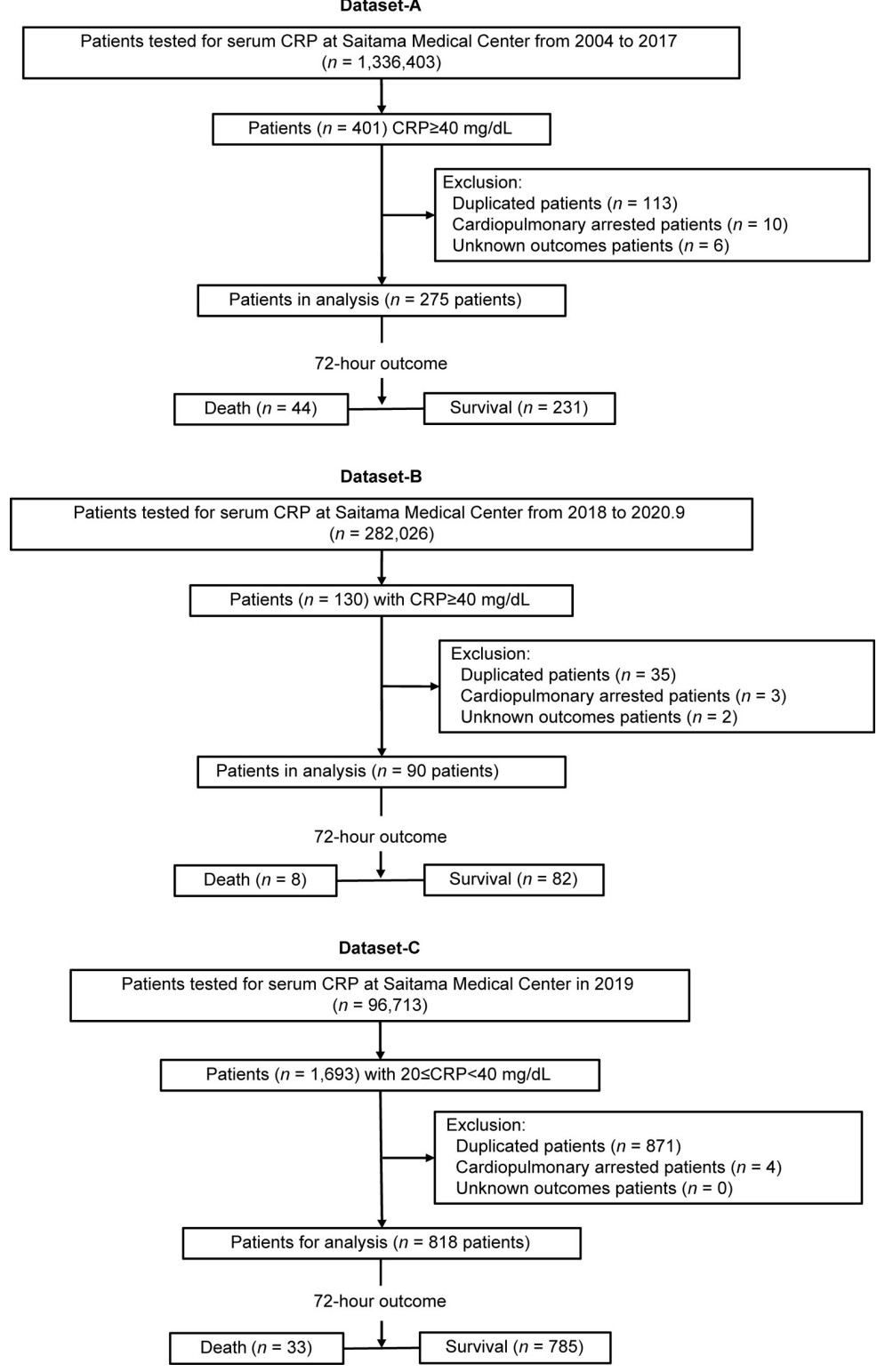

**Fig 1. Flow diagram outlining the patient selection.**

phosphate [IP], blood urea nitrogen [BUN], creatinine [Cre], uric acid, total cholesterol [TC], triglyceride, random plasma glucose, prothrombin time-international normalized ratio [PT-INR], activated partial thromboplastin time [APTT], fibrinogen [Fbg], D-dimer [DD], and antithrombin 3 [AT3]), total updated Charlson comorbidity index (CCI) scores [25], CCI components, and medications taken on the day of the CRP test. The following underlying causes of extremely high CRP were considered possible risk factors: sepsis, pneumonia, abscess, peritonitis, other infectious diseases, malignancy, cardiovascular disease, and gastrointestinal perforation.

## Statistical analysis

**Sample size.**   From a preliminary analysis, we obtained a dead survival ratio of 15:85 (mortality rate of 15%) among patients with CRP levels ≥40 mg/dL. Assuming a need to test the utility of a binary risk variable by assessing the proportions of the two groups, the sample size required to detect a difference of 0.20 in proportion was calculated as 211 (32 for the dead group vs. 179 for the surviving group) by setting a power of 80% and an alpha error of 5% [26]. As a result, we expanded the actual data size to 275 (with expected data sizes of 42 vs. 233 for the two groups) to ensure attainment of a higher power.

**Descriptive statistics.**   The summary values of all variables are presented as the median and inter-quartile range. The between-group differences were tested using Fisher's exact test for nominal variables and Mann–Whitney U test for numerical variables.

**Univariate analysis.**   The primary focus was to detect relevant risk variables associated with 72-hour mortality using receiver-operating characteristic (ROC) analysis and the Mann–Whitney test (or Fisher's exact test). The reliability of distinguishing dead from alive cases by each risk variable was expressed as the C-statistics (or area under the ROC curve) for the former and P-value for the latter.

**Multivariate logistic regression analysis.**   Multivariate logistic regression analysis (MLRA) was primarily used to build a regression model for predicting mortality risk.

$$p = \frac{1}{1 + e^{-X}} \; ; \; X = \beta_0 + \sum_{i=1}^{np} \beta_i x_i$$

where X represents a linear combination of risk variables ($x_i$) [I = 1~np], and $\beta_i$ is a partial regression coefficient for the i-th variable to be predicted by the maximum likelihood procedure. The 'p' represents a probability for belonging to the dead group to be calculated by assigning a set of variable xi (I = 1~np) from a given patient. The outcome of death (yes = 1, no = 0) 72 hours after the last CRP test was set as an object variable. The explanatory variables were all potential risk variables (demographic parameters, such as sex, age, BMI, primary disease states, and biochemical test results). The stepwise selection method was used to obtain an optimal combination of risk variables. For the laboratory tests, a distribution of values for each test was made approximately Gaussian by power transformation using the following Box–Cox formula [27].

$$X = \frac{x^\lambda - 1}{\lambda} \cdots (\lambda \neq 0.0); \; X = log(x) \cdots (\lambda = 0.0)$$

where $x$ and $X$ are a test result before and after the transformation, and λ is a power.

The power used for the major laboratory tests was $\lambda = 0.0$ (log-transformation) for LDH and DD, $\lambda = 0.3$ for Mg and BUN, $\lambda = 0.5$ for fibrinogen, and $\lambda = 0.7$ for K.

**Validation of the regression model predicted by MLRA.**   The actual flow of analyses were as follows: (1) To build a risk prediction model using the training **S1 Dataset** (275 records with CRP levels ≥40 mg/dL, composed of 44 dead and 231 alive cases); (2) to apply the

validation **S2 Dataset** (90 records with CRP levels ≥40 mg/dL, composed of 8 dead and 82 alive cases) to the regression model derived using **S1 Dataset**, and to calculate the predicted probability of all cases for belonging to the dead group; and (3) to calculate the reliability of prediction for both **S1 and S2 Datasets** as C-statistics in reference to the actual status of the mortality of each record.

For internal validation of the predicted regression model based on **S1 Dataset**, a boot-strap method [28], as used by repeated random re-sampling of n = 275 records in **S1 Dataset**, allowing duplicate sampling of the same data, and the reproducibility of the selected set of risk variables was evaluated.

In addition, the specificity of the regression model predicted by S1 Dataset from patients with CRP levels ≥40 mg/dL was assessed by comparison with a model predicted independently by **S3 Dataset** (818 records composed of 33 dead and 785 alive cases) from patients with $20 \leq CRP < 40$ mg/dL.

**Weighted average of risk scores (WARS).**   As an alternative method of risk prediction, we devised a risk index, called the weighted average of risk scores (WARS). The WARS is aimed at attaining flexibility and robustness in use by accommodating all available risk variables, even with partly missing data. The flow for the derivation of WARS is as follows:

1. For each risk variable, the degree of distinction between dead and alive cases was calculated as C-statistics using ROC analysis. For variables with a C-statistic >0.6, an optimal cutoff value for the distinction was determined as the point where the specificity equaled the sensitivity.

2. Each risk variable was graded by assigning a "weight" (wt) in reference to the boundaries of C-statistics, arbitrarily set at 0.65 and 0.70: wt = 1 (≤0.65), wt = 2 (0.65–0.7), and wt = 3 (>0.7), as shown in **Table 1**.

3. Binary transformation of each risk variable was performed (based on the respective cutoff values) to calculate the cumulative score. For example, when values of the dead group are shifted to a higher side, score 1 is assigned for any value above the cutoff; otherwise, the score is set to 0. Conversely, when the values of the dead group are shifted to a lower side, any value below the cutoff is assigned a score of 1.

4. The weighted average of risk score for each patient was calculated as

$$WARS = \frac{\sum_{i}^{np} wt_i \times sx_i}{\sum_{i}^{np} wt_i}$$

where $sx_i$ represents a score (0 or 1) assigned to a risk variable i in reference to the cutoff value, np represents the number of risk variables available for a given patient, and $wt_i$ represents the weight assigned to $sx_i$ (1, 2, or 3). Note that np can differ from one patient to another according to the number of missing results.

5. The utility of WARS for predicting the mortality risk was evaluated by univariate logistic regression analysis.

**Statistical software.**   The statistical package for StatFlex software version 7.0.11 (Artech Co. Ltd, Osaka, Japan) was used for data analysis, and G*Power version 3.1.9.4 [26] was used for sample size calculation.

**Table 1. Univariate comparison of laboratory data according to the outcome (dead/alive) at 72 hours.**

| | variables | Unit | n | Dead Me (IQR) [n] | Alive Me (IQR) [n] | ROC analysis | | | Wt *2 |
|---|---|---|---|---|---|---|---|---|---|
| | | | | | | C-statistic | cutoff *1 | P value by M-W | |
| ✓ | Age | year | 275 | 74 (62–79) [44] | 64 (52–72) [231] | **0.682** | **68** | **0.00013** | 2 |
| | RBC | 10⁴/μL | 275 | 342 (295–389) [44] | 363 (308–422) [231] | 0.567 | | 0.15295 | |
| | Hb | g/dL | 275 | 10.3 (8.8–12.2) [44] | 11.1 (9.3–12.8) [231] | 0.561 | | 0.19823 | |
| | TP | g/dL | 263 | 5.5 (4.9–6.0) [41] | 5.8 (5.1–6.5) [222] | 0.596 | | **0.04976** | |
| ✓ | Alb | g/dL | 264 | 2.0 (1.7–2.3) [42] | 2.5 (2.0–2.9) [222] | **0.686** | **2.2** | **0.00013** | 2 |
| | AST | U/L | 268 | 40 (25–64) [41] | 32 (19–63) [227] | 0.580 | | 0.09917 | |
| ✓ | LDH | U/L | 269 | 358 (236–750.0) [42] | 291 (200–417) [227] | **0.605** | **322** | **0.03126** | 1 |
| ✓ | K | mmol/L | 273 | 4.5 (4.00–5.5) [44] | 4.2 (3.6–4.6) [229] | **0.650** | **4.24** | **0.00151** | 1 |
| ✓ | IP | mg/dL | 219 | 4.7 (3.6–6.7) [38] | 3.4 (2.4–4.7) [181] | **0.699** | **4.0** | **0.00011** | 2 |
| ✓ | Mg | mg/dL | 95 | 2.7 (2.1–3.3) [10] | 2.1 (1.9–2.4) [85] | **0.747** | **2.28** | **0.01053** | 3 |
| ✓ | BUN | mg/dL | 275 | 59.5 (44.5–84.5) [44] | 35.0 (23.3–53.8) [231] | **0.732** | **48** | **0.00000** | 3 |
| | Cre | mg/dL | 274 | 1.8 (1.3–2.9) [43] | 1.3 (0.8–3.0) [231] | 0.561 | | 0.21428 | |
| | UA | mg/dL | 150 | 8.1 (4.95–9.75) [23] | 6.10 (4.53–8.30) [127] | **0.610** | | 0.09298 | |
| | TC | mg/dL | 79 | 108 (85.5–127.5) [8] | 127 (111–154) [71] | **0.690** | **120** | 0.07917 | |
| | Glu | mg/dL | 143 | 122 (88–171) [23] | 144 (120–209) [120] | **0.627** | | 0.05444 | |
| ✓ | Fbg | mg/dL | 91 | 538 (508–877) [13] | 896 (693–1108) [78] | **0.744** | **799** | **0.00509** | 3 |
| ✓ | D-dimer | μg/mL | 104 | 12.5 (7.7–28.7) [19] | 6.5 (3.8–13.8) [85] | **0.683** | **9.2** | **0.01277** | 2 |
| ✓ | CVD | 1 = yes; 0 = no | 275 | 1 (25%) vs. 0 (75%) [44] | 1 (8%) vs. 0 (92%) [231] | | | **0.00282** | 2 |

✓ indicates a candidate parameter adopted for use in the prediction modeling.

TC was omitted in the analysis because of small sample size.

*1 Cutoff values were determined as a test result where sensitivity = specificity, at the boundary of 0.65 and 0.70.

*2 Wt represents a weight for use in scoring.

Wt was graded into 1 to 3 based on C-statistics, Wt for CVD was arbitrary set to 2 from P<0.005.

C-statistics = area under the ROC curve (= AUC); RBC = red blood cells; Hb = hemoglobin; TP = total protein; Alb = albumin; AST = aspartate aminotransferase; LDH = lactate dehydrogenase; K = potassium; IP = inorganic phosphate; Mg = magnesium; BUN = blood urea nitrogen; Cre = creatinine; UA = uric acid; TC = total cholesterol; Glu = glucose; Fbg = fibrinogen; CVD = cardiovascular disease.

## Results

### Univariate analyses of demographics and risk variables

The characteristics of the study groups with respect to mortality, such as patient demographics, vital signs, laboratory test values, updated CCI, underlying causes of extremely high CRP, and medications are presented in **Tables 1 and S1**. The 72-hour mortality rate was 44/275 (16.0%), and the survival rate was 231/275 (84.0%). There was no significant difference in the CRP levels between the dead (44.1 mg/dL) and survival (43.7 mg/dL) groups. Detailed characterization of the two groups is presented in **Table 1** for variables that have the potential to be relevant for predicting mortality. The dead group had significantly higher values for age, LDH, K, IP, Mg, BUN, and DD than the survival group. Conversely, Alb, TC, and fibrinogen levels were significantly higher in the survival group. In **Table 1**, variables with C-statistics >0.6 are checkmarked at the head of respective rows, and their cutoff values for distinguishing the two groups are shown together with the weight (wt) for use in calculating the WARS.

In the subsequent analyses for building a risk prediction model using MLRA and calculating WARS, we selected 10 variables with C-statistics >0.6 or P < 0.01 by the Mann–Whitney test. TC was not included in calculating WARS due to the limited data (n = 79). The actual magnitude of between-group differences for these major variables is graphically shown in **Fig 2**.

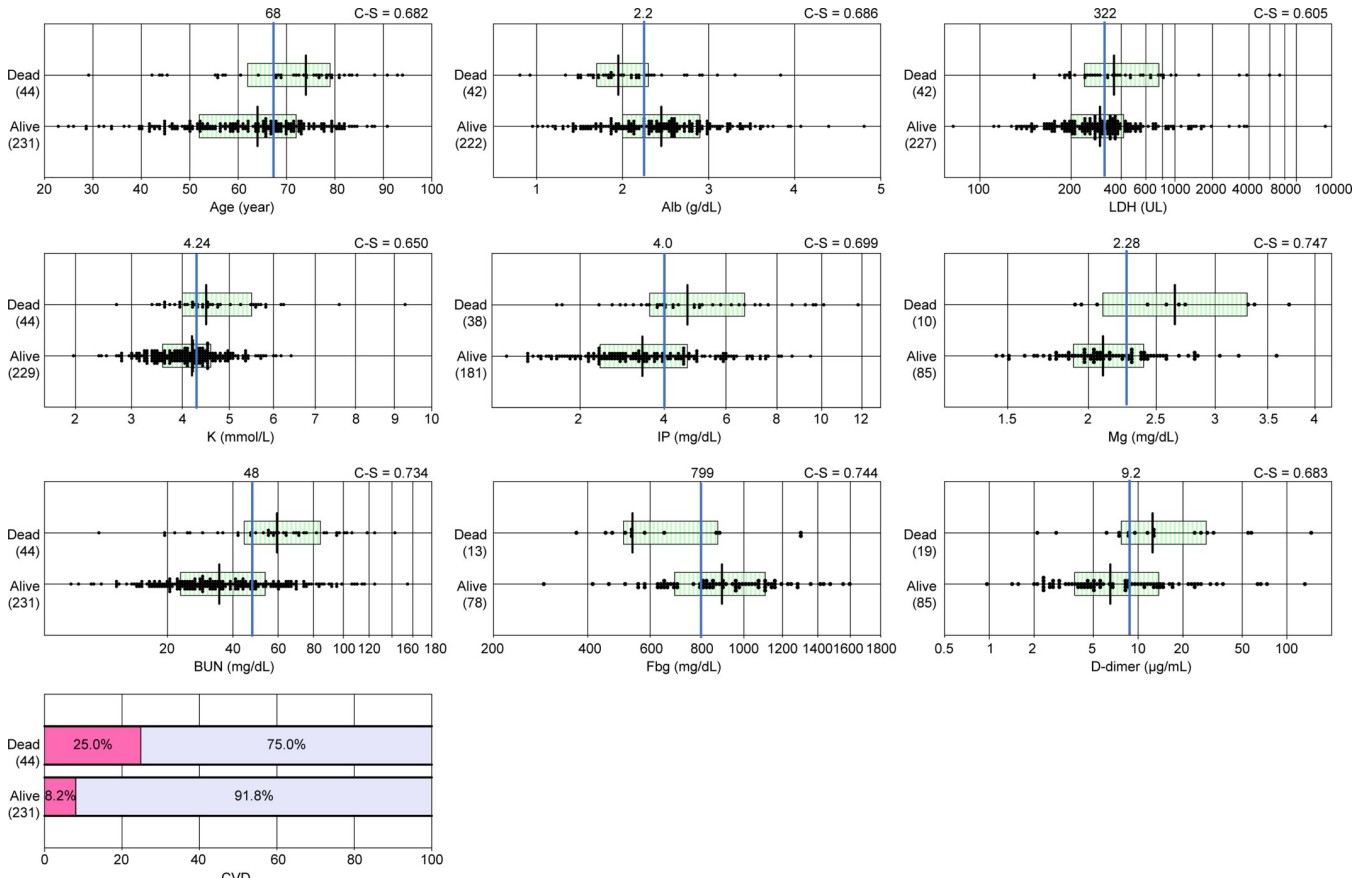

**Fig 2. Comparison of 10 major risk variables in relation to the outcome.** Patients with CRP levels ≥40 mg/dL (S1 Dataset) were divided by 72-hour outcome (44 dead and 218 alive). The utility of each risk variable for distinguishing the two groups was calculated as C-statistics (C-S) by ROC analysis. The cutoff value for the distinction (shown by the vertical line in the center) was determined as a value where the sensitivity is equal to the specificity.

## Prediction of mortality by MLRA and validation of the model

MLRA using S1 Dataset (n = 275; 44 dead, 231 alive) was performed to determine a regression model for predicting 72-hour mortality. An optimal regression model (A) was derived by stepwise selection of risk variables, which consisted of four risk variables: age, albumin, inorganic phosphate, and CVD as follows:

$$p = 1/[1 + \exp(-4.97 + 0.058(Age) - 0.835(ALB) + 0.615(IP^{0.5}) + 1.489(CVD)] \quad \text{(Eq 1)}$$

where '$p$' is the probability of a given patient belonging to the dead group. The accuracy of prediction in terms of C-statistics was 0.829 (95% CI = 0.760–0.900) (**Table 2**: Analysis-1).

For confirmation, the validation S2 Dataset (n = 90; 8 dead, 82 alive) was applied to the regression model (**Eq 1**), and the probability of belonging to the dead group (p) was computed for all 90 patients. The accuracy of the prediction based on $p$ values was calculated as 0.754 (95% CI = 0.595–0.913) in C-statistics with reference to the actual outcome: dead or alive (**Table 2**: Analysis-2). However, the C-statistics were not statistically different from those of S1 Dataset because of the wide 95% CI of the C-statistics as a result of the small sample size of S2 Dataset.

To test the specificity of the regression model for patients with CRP levels ≥40 mg/dL, S3 Dataset from patients with 20 ≤ CRP < 40 mg/dL (n = 818; 33 dead, 785 alive) was also

**Table 2. Multivariate logistic regression analyses for predicting 72-hour mortality.**

**Analysis (1) Dataset-A for training (CRP≧40: 2004~2017) n = 212 (with all 4 Exp Vars)**

| | | Exp Var | β | SE(β) | z | P | OR | 95%CI |
|---|---|---|---|---|---|---|---|---|
| | 0 | | -4.969 | 1.666 | | | | |
| | 1 | Age | 0.058 | 0.0187 | 3.113 | 0.0019 | 1.06 | 1.022–1.099 |
| | 2 | Alb | -0.835 | 0.3659 | -2.282 | 0.0225 | 0.434 | 0.212–0.889 |
| | 3 | IP | 0.615 | 0.2207 | 2.788 | 0.0053 | 1.85 | 1.201–2.852 |
| | 4 | CVD | 1.489 | 0.6146 | 2.422 | 0.0154 | 4.431 | 1.328–14.78 |

MLRA: Obj Var = Death

AIC = 158.588, C-S = 0.829 (95%CI = 0.759–0.900)

$$P = 1/[1+\exp(-4.97 + 0.058(Age) - 0.835(ALB) + 0.615(IP^{0.5}) + 1.489(CVD)]$$

**Analysis (2) Dataset-B for validation (CRP≧40: 2018~2020) n = 73 (with all 4 Exp Vars)**

ROC analysis to evaluate the accuracy of predicted probability ($p$) for the fatal outcome.

| N dead | N alive | C-S | 95%CI of C-S |
|---|---|---|---|
| 8 | 65 | 0.754 | 0.595–0.913 |

**Analysis (3) Dataset-C for testing specificity (20≦CRP<40: 2019) n = 818 (with all 4 Exp Vars)**

ROC analysis to evaluate the accuracy of predicted probability ($p$) for the fatal outcome.

| N dead | N alive | C-S | 95%CI of C-S |
|---|---|---|---|
| 33 | 785 | **0.788** | 0.695–0.881 |

**Analysis (4): Dataset-C for testing specificity (20≦CRP<40: 2019) n = 720 (with all 5 Exp Vars)**

| | | Exp Var | β | SE(β) | z | P | OR | 95%CI |
|---|---|---|---|---|---|---|---|---|
| | 0 | | -8.547 | 1.803 | | | | |
| | 1 | Alb | -1.343 | 0.4262 | -3.151 | **0.0016** | 0.261 | 0.113–0.602 |
| | 2 | BUN | 0.492 | 0.1134 | 4.339 | **0.0000** | 1.636 | 1.310–2.043 |
| | 3 | LDH | 0.572 | 0.2129 | 2.685 | **0.0073** | 1.771 | 1.167–2.688 |
| | 4 | Ht | 0.365 | 0.1237 | 2.948 | **0.0032** | 1.44 | 1.130–1.84 |
| | 5 | Hb | -0.918 | 0.3843 | -2.389 | **0.0169** | 0.399 | 0.188–0.848 |

MLRA: Obj Var = Death

AIC = 182.899, C-S = 0.906 (95%CI = 0.867–0.943)

$$p = 1/[1+\exp(-8.55-1.343(ALB) + \cdots\cdots - 0.918(Hb)]$$

Obj Var = object variable; Exp Var = explanatory variable; β = partial regression coefficient; SE = standard error; OR = odds ratio; C-S = C-statistics; ALB = albumin; BUN = blood urea nitrogen; IP = inorganic phosphate; CVD = cardiovascular diseases; ALB = albumin; BUN = blood urea nitrogen; IP = inorganic phosphate; Ht = haematocrit; Hb = haemoglobin.

applied to **Eq 1** to compute the probability of belonging to the fatal outcome (p). The accuracy of the prediction was 0.788 (95% CI = 0.695–0.881) (**Table 2**: Analysis-3). This reduction from 0.829 was not statistically significant since 0.829 is within the 95% CI of C-statistics for S3 Dataset. Independent derivation of a mortality prediction model was performed by using S3 Dataset through stepwise selection of risk variables. The regression model was quite different from **Eq 1** as shown in **Table 2**: Analysis-4. In this case, Alb was chosen again, age and IP were not selected, and BUN, LDH, Ht, and Hb were newly included in the model.

As a work of internal validation with respect to the reproducibility of **Eq 1**, bootstrap resampling of S1 Dataset, which allows for replacement (duplicate sampling of the same data), was conducted 25 times, and a risk prediction model was derived by MLRA for each dataset. Three variables, Mg, Fbg, and DD, were not included in the bootstrap analysis due to limited data. The results shown in **Table 3** demonstrate that a certain degree of variability is inevitable in the automatic selection of risk variables, except for age.

Table 3. Reproducibility of the prediction model by MLRA.

| Rep | N | AUC | Age | Alb | LDH | K | IP | BUN | CVD |
|---|---|---|---|---|---|---|---|---|---|
| Original | 212 | 0.829 | ◎ | ○ |  |  | ◎ |  | ○ |
| 1 | 201 | 0.846 | ◎ |  | ◎ |  | ◎ |  |  |
| 2 | 245 | 0.843 | ◎ | ◎ | ○ | ◎ |  |  | ○ |
| 3 | 220 | 0.828 | ◎ |  |  |  | ◎ |  | ○ |
| 4 | 220 | 0.879 | ◎ | ◎ |  |  | ○ |  | ◎ |
| 5 | 222 | 0.859 | ◎ |  |  |  | ◎ |  | ◎ |
| 6 | 275 | 0.832 | ◎ |  |  |  |  | ◎ | ◎ |
| 7 | 260 | 0.832 | ◎ | ◎ |  |  |  |  |  |
| 8 | 207 | 0.875 | ◎ | ◎ |  |  | ◎ |  | ○ |
| 9 | 264 | 0.869 | ◎ | ◎ | ◎ |  |  | ○ | ○ |
| 10 | 263 | 0.876 | ◎ | ◎ | ○ | ◎ |  |  |  |
| 11 | 218 | 0.86 | ◎ | ○ |  |  | ○ |  | ◎ |
| 12 | 215 | 0.892 | ◎ |  |  |  | ◎ |  | ◎ |
| 13 | 270 | 0.825 | ◎ |  | ○ | ○ |  |  | ◎ |
| 14 | 214 | 0.827 | ◎ | ◎ |  |  | ○ |  |  |
| 15 | 220 | 0.867 | ◎ |  |  |  | ◎ |  | ◎ |
| 16 | 225 | 0.85 | ◎ |  |  |  | ◎ |  | ◎ |
| 17 | 224 | 0.896 | ◎ |  |  |  | ◎ |  | ○ |
| 18 | 231 | 0.786 | ◎ |  |  |  | ◎ |  |  |
| 19 | 192 | 0.85 | ◎ | ◎ | ○ |  | ○ |  |  |
| 20 | 263 | 0.863 | ◎ | ○ |  |  |  | ◎ | ◎ |
| 21 | 224 | 0.866 | ◎ | ◎ |  |  | ◎ |  |  |
| 22 | 218 | 0.85 | ◎ | ◎ | ◎ |  | ◎ |  |  |
| 23 | 252 | 0.849 | ◎ | ◎ | ◎ |  |  |  |  |
| 24 | 275 | 0.825 | ◎ |  |  |  |  | ◎ | ◎ |
| 25 | 223 | 0.81 | ◎ | ○ |  |  | ○ |  |  |

◎: P < 0.001; ○: P < 0.01 Rep = iteration numver; N = valid sample size; AUC = area under the curve; Alb = albumin; LDH = lactate dehydrogenase; K = potassium; IP = inorganic phosphate; BUN = blood urea antigen; CVD = cardio vascular disease.

## Utility of WARS for the prediction of mortality

The WARS was calculated from S1 Dataset by combining the risk scores of 10 major risk variables, which are shown in Fig 2, using the following formula:

$$WARS = \{(bK) + (bLDH)\} + 2\{(bAge) + (bALB) + (bIP) + (bDD) + (CVD)\} + 3\{(bMg) + (bBUN) + (bFbg)\} / \sum_{i=1}^{np} wt_i$$

where prefix 'b' implies the binary transformation of original variables. Note that the denominator changes from one patient to another according to the number of missing data.

Univariate logistic regression analysis was performed to derive a risk prediction model, as shown in **Table 4**.

$$p = 1/[1 + \exp(-4.42 + 5.954 \times WARS)] \tag{Eq 2}$$

The probability (p) for belonging to the dead group was calculated for all 275 cases, and the accuracy of the prediction was determined as C-statistics of 0.837 (95% CI: 0.780–0.893); this was a slight improvement from that of the MLRA model (0.829), although the difference was

**Table 4. Accuracy of WARS computed from 10 risk variables for predicting 72-hour mortality.**

| Analysis (1): Dataset-A (CRP ≧ 40: 2004~2017) for deriving a risk prediction model using WARS | | | | | | | |
|---|---|---|---|---|---|---|---|
| WARS; Obj Var = Death    n = 275 | | | | | | | |
| | Exp Var | β | SE(β) | z | P | OR | 95% CI |
| 0 | | -4.417 | 0.5355 | | | | |
| 1 | WARS | 5.954 | 0.9375 | 6.35 | 3.11E-10 | 385.2 | 61.3–2419.2 |
| AIC = 193.63, C-S = 0.837 (95% CI = 0.780–0.893), E = exponential | | | | | | | |
| | $p = 1/[1+\exp(-4.42 + 5.954 \times WARS)]$ | | | | | | |
| **Analysis (2): Dataset-B (CRP ≧ 40: 2018~2020) for validation of WARS. n = 90** | | | | | | | |
| ROC analysis to evaluate the accuracy of predicted probability (p) for the fatal outcome. | | | | | | | |
| N dead | N alive | C-S | 95%CI of C-S | | | | |
| 8 | 82 | **0.745** | 0.546–0.944 | | | | |
| **Analysis (3): Dataset-C (20≦CRP<40: 2019) for testing specificity of WARS. n = 818** | | | | | | | |
| ROC analysis to evaluate the accuracy of predicted probability (p) for the fatal outcome. | | | | | | | |
| N dead | N alive | C-S | 95%CI of C-S | | | | |
| 33 | 785 | **0.785** | 0.705–0.865 | | | | |

Obj Var = object variable; Exp Var = explanatory variable; β = partial regression coefficient; SE = standard error; OR = odds ratio; C-S = C-statistics; E-10 = $10^{-10}$. WARS = weighted average risk score.

not statistically significant. Applying validation S2 and S3 Datasets to **Eq 2** resulted in C-statistics of 0.745 (0.546–0.944) and 0.785 (0.705–0.865), respectively.

**Fig 3** displays the comparison of utility among mortality risk variables by ROC analysis with MLRA and WARS.

## Discussion

### Pathological and prognostic implications of extremely high serum CRP

CRP is an expedient inflammatory biomarker with rapid kinetics in response to inflammation. The association between prognosis and high CRP has been reported in several diseases, including septic shock [29], bacterial infection [30], acute ischemic stroke [31], acute idiopathic pericarditis [32], unstable angina or non-Q-wave myocardial infarction [33], and most adult solid tumors [34], including advanced non-small cell lung cancer [35], urothelial cancer along with renal cell carcinoma, prostate cancer, bladder cancer, and upper urinary tract urothelial carcinoma [36]. Furthermore, a recent study demonstrated significant differences in CRP levels between patients who died and those who survived following 48 h and 96 h after admission to a palliative care unit [37], as well as patients who were hospitalized in internal medicine wards [38]. Regarding the critical level of CRP, several reports regard CRP levels ≥40 mg/dL as a panic value with imminent mortality risk [15–17]. Moreover, the cutoff value of CRP ≥40 mg/dL is often used to predict sepsis in emergency departments [19], as well as treatment failure among patients with exacerbated mild to moderate chronic obstructive pulmonary disease [20, 21]. In contrast, Silvestre et al. suggested that CRP levels ≥40 mg/dL were not associated with increased mortality among patients with sepsis during an ICU stay [14]. Accordingly, the prognostic implication of CRP levels ≥40 mg/dL remains controversial, and it is of clinical importance to elucidate mortality risk variables and devise a numerical model for predicting short-term mortality in patients with extremely high CRP level.

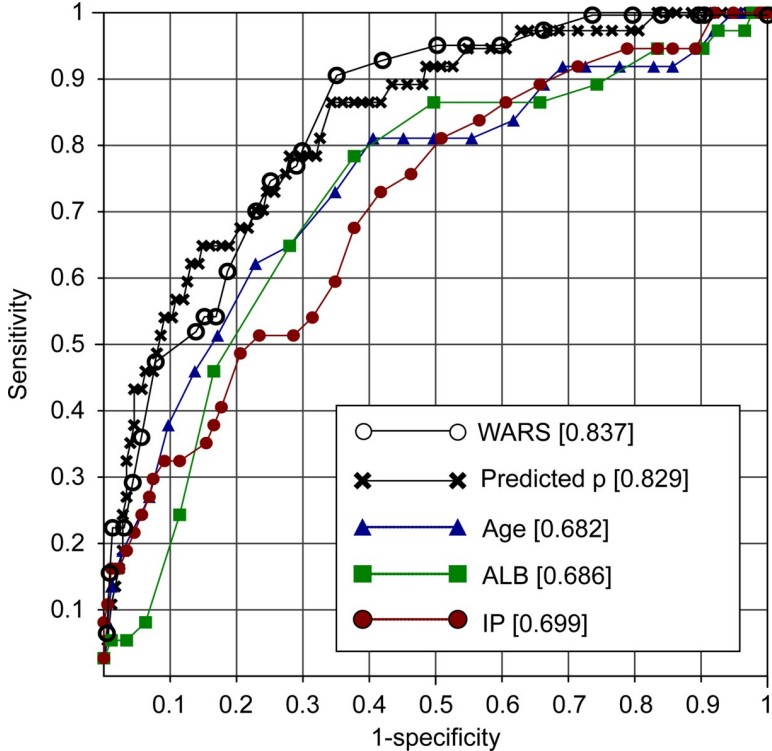

**Fig 3. Comparison of the utility among mortality risk variables by ROC analysis.** ROC analysis was performed using S1 Dataset to compare the utility of the following mortality risk variables: predicted probability (p) derived by MLRA, WARS, serum albumin, and serum IP. The C-statistics calculated for each variable is shown attached to its name.

## Study summary

In the current study, we explored mortality risk factors in patients with extremely high CRP levels. The 72-hour mortality rate among 275 patients with CRP levels ≥40 mg/dL was 16.0%, but there was no significant difference in CRP levels between the dead and surviving groups. Using MLRA, we revealed four main independent risk variables as follows: age, serum Alb, serum IP, and the CVD status. However, internal validation of MLRA results demonstrated that there were other relevant variables in predicting mortality, namely, BUN, K, Mg, LDH, fibrinogen, and DD, as shown in **Table 3**.

## Mortality risk variables and their pathophysiological implications

Serum albumin concentration is closely correlated with health homeostasis [39]. The median serum Alb in the dead group was 2.0 g/dL; hence, the group was considered extremely mal-nourished [40]. Many studies have established that hypoalbuminemia is one of the strongest prognostic factors related to mortality [41, 42], morbidity [41], and length of hospitalization [41], including mortality in elderly patients [43]. Therefore, an extremely low Alb level is well established as a mortality risk.

The normal range of serum IP is 2.5.4.5 mg/dL. In this study, the median serum IP value of the dead group was 4.7 mg/dL, in contrast to 3.4 mg/dL in the surviving group. It has been reported that a serum IP level >5.5 mg/dL is associated with an increase in the risk of cardio-vascular and all-cause mortality in patients with mild or moderate renal function impairment

[44]. Indeed, Dhingra et al. reported a strong relationship between all-cause mortality for 45 months in patients with chronic kidney disease and serum IP concentration [45]. Moreover, high serum IP levels have been shown to be associated with the risk of cardiovascular disease and high mortality [45]. Naffa et al. reported that elevated serum IP levels affect the mortality rate of patients with community-acquired pneumonia [46]. Another study in our hospital targeted patients with extremely high AST, and demonstrated that serum IP was an excellent predictor of short-term prognosis [47]. We postulate that there are several mechanisms that lead to elevated IP, including renal failure with decreased reabsorption of IP from renal tubules, leakage from apoptotic cells, and intense muscular exertion in near-death agony.

Regarding the other risk variables that were identified by univariate analysis, BUN is also well documented as a mortality risk factor [48, 49]. Increased BUN during the period of impending death is caused by dehydration, renal dysfunction, or enhanced catabolism, which lead to the breakdown of proteins with increased tissue release of ammonia. Hyperkalemia prior to death can be explained by renal failure, apoptotic release from cells during multi-organ failure, or increased intravascular hemolysis. Hypermagnesemia is also known to occur in renal and/or cardiac failure, and is associated with imminent death. Furthermore, increased serum LDH is commonly encountered in patients with impending death due to hypoxemia in circulation failure, leakage from damaged cells in the liver, muscles, and erythrocytes, or end-stage cancer tissue. The relative decrease in fibrinogen and increased DD prior to death can be explained by progressive occlusion of small vessels caused by hypotension and hypoperfusion.

## Limitations of MLRA in predicting mortality risk

In summary, the heterogeneous variables identified in this study are relevant in predicting mortality regardless of the type of primary disease. However, it is not feasible to reflect all of these variables together in building a regression model because the method does not allow mutually correlated variables to be included together. Therefore, the selection of variables in the lower order of importance can change easily with a slight change in the dataset, as illustrated in Table 3. Another problem we encountered in performing MLRA was that the regression model does not allow any missing data, and thus, risk variables that were measured less frequently tended to not be included in the model even with high C-statistics for predicting mortality; in other words, MLRA ignores patients with missing data that are required in the regression model. This is another drawback when working with a heterogeneous set of clinical records, which inevitably contain a large amount of missing data.

## WARS as an alternative method of predicting mortality risk

From this perspective, we propose to use WARS as an alternative method of predicting mortality because it can cover a wider range of risk variables even with partly missing data. In the current study, the performance of WARS in mortality prediction was found to be comparable or superior to that of MLRA.

However, WARS must be performed with caution because, unlike MLRA it does not consider correlations among the risk variables. Therefore, if risk variables have very high correlations with each other, such as BMI and body fat%, or AST and ALT. Inclusion of these factors together in calculating WARS may factitiously increase the risk score. A confounding phenomenon has the potential to be problematic when using WARS. For example, assuming that age is a primary risk variable and the two groups to be distinguished show differences in age, a risk variable that is influenced by age tends to have a higher weight in the calculation of WARS. Fortunately, we did not detect such a phenomenon in computing WARS from S1 Dataset.

It is of note that WARS can be readily applicable to any clinical situation that requires a numerical model for predicting a prognosis among a cohort of patients based on a heterogeneous combination of clinical and laboratory findings, even in the presence of many missing results using the following procedures: (1) To perform univariate ROC analysis of prognosis with derivation of a cutoff value and C-statistics, (2) to dichotomize or 0–1 scoring of each variable for prediction in reference to the cutoff value, (3) assign appropriate weight to each variable based on the C-statistics, and (4) compute WARS as a weighted average score of risk variable chosen for use in predicting the prognosis.

## Limitations

The current study has several limitations. First, this was a single center retrospective study conducted at Saitama Medical Center in Japan, and our findings may not be generalizable to other populations. Second, we selected patients with CRP levels over 40 mg/dL (only 0.0302% of our cohort); therefore, the clinical relevance of our findings for the general population is limited. In fact, the risk factors for 72-hour mortality identified by MLRA were different in patients with lower CRP levels, although WARS was still applicable to them in predicting mortality due to the wider coverage of risk variables.

## Conclusion

We aimed to develop a numerical model for predicting 72-hour mortality in patients with a disputed CRP level of ≥40 mg. Multiple risk variables were identified by univariate analysis, including increased age, IP, BUN, Mg, K, LDH, DD, low Alb and fibrinogen, and presence of CVD. By MLRA, the selected variables in the optimal regression model were limited to age, Alb, IP, and CVD; this was due to similar predictability among other variables and the presence of many missing data in some variables. In contrast, mortality prediction using WARS resulted in a performance comparable to that of MLRA, and exhibited a superior property of covering all relevant risk variables and robustness in use even with missing data. Therefore, we believe that the novel WARS should be used as the method of choice for predicting 72-hour mortality in patients with extremely high CRP levels. WARS has the potential to assist physicians in decision-making, provide therapeutic and management options to patients and their families, and improve the quality of initial medical management in both primary and critical care settings.

## Supporting information

**S1 Table. Patients' medications.** ACEI: Angiotensin-converting-enzyme inhibitor, ARB: Angiotensin II receptor blocker, CCB: Calcium channel blocker, NSAID: Nonsteroidal anti-inflammatory agent, PPI: Proton pump inhibitors, HAART: Highly active antiretroviral therapy. *P*-values were calculated using Fisher's exact test.
(DOCX)

**S1 Dataset.**
(XLSX)

**S2 Dataset.**
(XLSX)

**S3 Dataset.**
(XLSX)

## Acknowledgments

A summary of this study was presented at the Beginning of Medical Students and Residents for the Japanese Society of Internal Medicine at the 116[th] Annual Meeting of the Japanese Society of Internal Medicine (Nagoya, April 27, 2019). Editorial support, in the form of medical writing, assembling tables, and creating high-resolution images based on the authors' detailed directions, collating author comments, copyediting, fact-checking, and referencing, was provided by Editage, Cactus Communications.

## Author Contributions

**Conceptualization:** Kai Saito, Hitoshi Sugawara, Kiyoshi Ichihara.

**Data curation:** Kai Saito, Hitoshi Sugawara.

**Formal analysis:** Kai Saito, Hitoshi Sugawara, Kiyoshi Ichihara.

**Funding acquisition:** Hitoshi Sugawara.

**Investigation:** Kai Saito, Hitoshi Sugawara.

**Supervision:** Hitoshi Sugawara, Kiyoshi Ichihara, Tamami Watanabe, Akira Ishii, Takahiko Fukuchi.

**Writing – original draft:** Kai Saito, Hitoshi Sugawara.

**Writing – review & editing:** Hitoshi Sugawara, Kiyoshi Ichihara, Tamami Watanabe, Akira Ishii, Takahiko Fukuchi.

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
