## [Decision Letter · Decision Letter 0]

21 May 2020

PONE-D-20-07042

Predicting 72-hour mortality in patients with extremely high serum C-reactive protein levels over 40 mg/dL: a case-controlled cross-sectional study.

PLOS ONE

Dear Dr. Sugawara,

Thank you for submitting your manuscript to PLOS ONE. After careful consideration, we feel that it has merit but does not fully meet PLOS ONE’s publication criteria as it currently stands. Therefore, we invite you to submit a revised version of the manuscript that addresses the points raised during the review process.

The reviewers consider that the size of the study should be justified further. The inclusion of an additional group of individuals with intermediate CRP levels is an interesting suggestion. Statistical treatment of the data has to be revised.

We would appreciate receiving your revised manuscript by Jul 05 2020 11:59PM. To enhance the reproducibility of your results, we recommend that if applicable you deposit your laboratory protocols in protocols.io, where a protocol can be assigned its own identifier (DOI) such that it can be cited independently in the future. For instructions see: http://journals.plos.org/plosone/s/submission-guidelines#loc-laboratory-protocols

We look forward to receiving your revised manuscript.

Kind regards,

Pablo Garcia de Frutos

Academic Editor

PLOS ONE

Journal Requirements:

2. In the ethics statement in the manuscript and in the online submission form, please provide additional information about the patient records used in your retrospective study. Specifically, please ensure that you have discussed whether all data were fully anonymized before you accessed them and the name of the committee that waived the requirement for informed consent. If patients provided informed written consent to have data from their medical records used in research, please include this information.

Additional Editor Comments (if provided):

Reviewers' comments:

Reviewer's Responses to Questions

**Comments to the Author**

1. Is the manuscript technically sound, and do the data support the conclusions?

Reviewer #1: Yes

Reviewer #2: Yes

2. Has the statistical analysis been performed appropriately and rigorously? 

Reviewer #1: Yes

Reviewer #2: No

3. Have the authors made all data underlying the findings in their manuscript fully available?

Reviewer #1: No

Reviewer #2: Yes

4. Is the manuscript presented in an intelligible fashion and written in standard English?

Reviewer #1: Yes

Reviewer #2: Yes

5. Review Comments to the Author

Reviewer #1: The authors present an interesting manuscript entitled, "Predicting 72-hour mortality in patients with extremely high serum C-reactive protein levels over 40 mg/dL: a case-controlled cross-sectional study".

However, I would like to have seen the authors either:

(1) Validating the findings by seeing whether the model can accurately predict patient survival outcomes observed in a second set of retrospective data (possibly acquired from another hospital or a different time period) to provide convincing evidence for the value of the prediction model.

…or, if access to other CRP data is not possible:

(2) Providing direct comparison with patients with high CRP but below the 40 mg/dL threshold. By comparing against another subgroup within your own dataset with high (but not extremely high) CRP and otherwise matched exclusion criteria would have confirmed whether the prediction model breaks down (changes substantially) at lower CRP thresholds and this might have highlighted differing risk factors amongst differing groups.

General feedback on the manuscript is as follows…..

The title and abstract were appropriate.

The introduction was reasonable but the authors need to introduce the concept and implications of extremely high serum CRP in clinical settings and frequency in patients. The authors only note CRP levels in healthy subjects and how it can become elevated 10,000-fold in hospital patients.

The methods section is robust for most part but there is a large contradiction in the section named “Study design and participant selection” with regards the number of patients included in the study (lines 95-100 contradict with lines 103-108). Also, the occurrence of 0.1% (as stated by the authors) suggests there is only 283 adult patients with extremely high CRP out of a total number of 283,000 adult patients over 18 years old. If so, I presume most of the CRP data list (i.e. the remaining 625, 750 patients) was not useful/relevant? This statement needs clarification to explain how often an adult patient (above 18 years old) had a CRP level over 40 mg/dL. Lastly, the authors should refer the reader to the Tables when stating ‘underlying causes of extremely high CRP’ (lines 123-124) were considered; otherwise those lines are somewhat vague.

The results section was good for the most part, except the statement on lines 204-205 seems confused to me and it is not clear why potassium levels were omitted (not considered further) in the Prediction Model (lines 208 onwards).

The limitations need to note that the findings only constitute a very small cohort of patients (especially after applying your exclusion criteria) from the pool of typical clinical CRP samples taken in hospital so the impact/relevance is relatively low & limited despite being of medical importance to affected individuals.

The discussion is reasonable throughout but the authors have neglected to discuss some important findings of the study, including those relating to dementia and significant data relating to biochemical measurements such as potassium, albumin and total cholesterol levels.

The conclusion is appropriate for the most part but to avoid overplaying the findings in lines 293-295 the authors need to reiterate that the utility of the proposed prediction model to enable physicians to easily estimate the 72-hour mortality probability and facilitate an easier decision-making process only applies to adult patients falling outside your designated list of exclusion criteria.

There were some minor typological and spelling errors noted (see file attached).

Reviewer #2: CRP is a well-established marker of systemic inflammation. Because the levels of CRP rise much more significantly during acute inflammation than the levels of the other acute phase reactants, the CRP test has been used for decades to indicate the presence of systemic inflammation, infection, or sepsis.

The present study is the hospital-based study to investigate the risk factors associated with 72-hour mortality in people with extremely high CRP. The main findings were that the 72-hour mortality rate in patients with extremely high CRP was 14.2% and that the main independent risk factors associated with mortality were age, serum inorganic phosphorus, and blood urea nitrogen. level.

The manuscript is interesting, but suffers with some limitations. The main questions to the authors are:

1. It is not clear why the value of CRP over 40 mg/dl was set as cut-off? According to Póvoa the CRP levels in patients with sepsis, severe sepsis and septic shock were 15.2 ± 8.2, 20.3 ± 10.9 and 23.3 ± 8.7 mg/dL. [Póvoa P, et al. C-reactive protein as a marker of infection in critically ill patients. Clin Microbiol Infect. 2005]. Moreover, according to the reference 22 (Line 96) high value of CRP is not a good marker of the mortality rate prognosis. The ICU mortality rate of septic patients with CRP < 10, 10-20, 20-30, 30-40 and >40 mg/dL was 20, 34, 30.8, 42.3 and 39.1%, respectively (P = 0.7). So there is no reason to choose the cut-off value of 40 mg/dl.

2. It would be important to have more baseline clinical data about the patient population (e.g. medication).

3. Small sample size. Authors calculated the required sample size as 23 deaths and 138 survivals to achieve a power of 80% and an alpha error of 5%. The 72-hour mortality in the investigation was 26 and three variables were included into the multivariate logistic regression analysis. It is recommended to use at least 10 events per variable to fit prediction models in clustered data using logistic regression. Up to 50 events per variable may be needed when variable selection is performed. [Wynants L, et al. A simulation study of sample size demonstrated the importance of the number of events per variable to develop prediction models in clustered data. J Clin Epidemiol. 2015 Dec;68(12):1406-14.]. Age (years), serum inorganic phosphorus (mg/dL) and blood urea nitrogen value (mg/dL) are continuous variables. Into the investigation were included only 26 events. So the power of the investigation is not sufficient.

6. PLOS authors have the option to publish the peer review history of their article (what does this mean?). If published, this will include your full peer review and any attached files.

Reviewer #1: No

Reviewer #2: No

---

## [Author Response · Author response to Decision Letter 0]

19 Dec 2020

Response to Reviewer’s Comments

Journal Requirements:

 Please ensure that your manuscript meets PLOS ONE's style requirements, including those for file naming. The PLOS ONE style templates can be found at https://journals.plos.org/plosone/s/file?id=wjVg/PLOSOne_formatting_sample_main_body.pdf and https://journals.plos.org/plosone/s/file?id=ba62/PLOSOne_formatting_sample_title_authors_affiliations.pdf

Our response→ We have ensured that our manuscript meets PLOS ONE´s requirements.

 In the ethics statement in the manuscript and in the online submission form, please provide additional information about the patient records used in your retrospective study. Specifically, please ensure that you have discussed whether all data were fully anonymized before you accessed them and the name of the committee that waived the requirement for informed consent. If patients provided informed written consent to have data from their medical records used in research, please include this information. 

Our response→ The study protocol was designed according to the tenets of the Declaration of Helsinki and approved by the Institutional Clinical Ethics Review Boards of Saitama Medical Center, Jichi Medical University, Saitama, Japan (Clinical #10-79 and #S20-025). We ensured that the data from the patients’ records were fully anonymized. The requirement for informed written consent was waived due to the retrospective nature of this study.

 We note that you have indicated that data from this study are available upon request. PLOS only allows data to be available upon request if there are legal or ethical restrictions on sharing data publicly. For information on unacceptable data access restrictions, please see http://journals.plos.org/plosone/s/data-availability#loc-unacceptable-data-access-restrictions. In your revised cover letter, please address the following prompts: 

b) If there are no restrictions, please upload the minimal anonymized data set necessary to replicate your study findings as either Supporting Information files or to a stable, public repository and provide us with the relevant URLs, DOIs, or accession numbers. Please see http://www.bmj.com/content/340/bmj.c181.long for guidelines on how to de-identify and prepare clinical data for publication. For a list of acceptable repositories, please see http://journals.plos.org/plosone/s/data-availability#loc- recommended-repositories. 

Our response→ The study protocol was approved by the Institutional Clinical Ethics Review Boards (IRB) and did not include the availability of public data sharing. The data does not include any potentially identifying or sensitive patient information. The data are available to interested researchers upon reasonable request to the corresponding author and will be provided once approved by the IRB.

Review Comments to the Author

Our response→ Dr. Kiyoshi Ichihara has joined the study and was included in the manuscript as a co-author. We have extensively revised the manuscript throughout, according to the comments of Reviewer #1 and Reviewer #2. Newly added tests are indicated in dark blue font, while dark red font is used to indicate partly corrected phrases or numerical values. Please allow us to delete large blocks of texts without leaving track changes for readability. 

Reviewer #1: The authors present an interesting manuscript entitled, "Predicting 72-hour mortality in patients with extremely high serum C-reactive protein levels over 40 mg/dL: a case-controlled cross-sectional study"

Our response→ We are very grateful for the detailed review of our manuscript and the invaluable comments from the reviewers. In an attempt to revise the manuscript, we realized that support from a biostatistician was necessary to cope with the issues raised. Fortunately, we obtained support from Prof. Kiyoshi Ichihara, a biostatistician with a medical background, with a particular focus on laboratory medicine. With his help, we not only tried to validate our methodology, but also devised an alternative way of predicting the mortality of patients with extremely high CRP levels. 

The details of the changes are described below in response to the comments. We apologize for the lengthy explanations and appreciate your time in reviewing our responses. We hope that they are comprehensive and scientifically acceptable. 

However, I would like to have seen the authors either:

(1) Validating the findings by seeing whether the model can accurately predict patient survival outcomes observed in a second set of retrospective data (possibly acquired from another hospital or a different time period) to provide convincing evidence for the value of the prediction model.

Our response→ According to the reviewer’s suggestion, we first extended the study period for 3 years (from 2004–2014 to 2004–2017) to increase the precision of the analyses. Therefore, the total sample size increased from 183 to 275, while the deceased cases increased from 28 to 44 cases). We regarded this as a training dataset (Dataset-A) and made an additional attempt to further extend the period by 2.5 years (Jan 2018 – Sep 2020) to obtain a validation dataset (Dataset-B), which was composed of 90 cases (8 dead and 72 alive).

As a result, we found that the accuracy of the predicting death in terms of C-statistics was 0.829 (slightly decreased from 0.868), and the training dataset gave an accuracy of 0.754. Although we admit that the accuracy was decreased, we understand that it is within an allowable limit of variation in consideration of the small sample size (n = 90) (Table 2, Analysis 1 and 2)

…or, if access to other CRP data is not possible:

(2) Providing direct comparison with patients with high CRP but below the 40 mg/dL threshold. By comparing against another subgroup within your own dataset with high (but not extremely high) CRP and otherwise matched exclusion criteria would have confirmed whether the prediction model breaks down (changes substantially) at lower CRP thresholds and this might have highlighted differing risk factors amongst differing groups.

Our response→ We attempted this following the suggestion of the reviewer. To this end, we newly retrieved the dataset (Dataset-C) with CRP values between 20 and 40 for 1 year (2019), which amounted to 818 cases (33 deceased and 785 alive). Using the prediction formula prepared from the training dataset, the accuracy of prediction in C-statistics was 0.788. Therefore, we considered that the prediction formula was also applicable to the lower CRP group, but with a compromised power of prediction (Table 2, Analysis 3)

To assess the implication of this finding, we performed independent multivariate logistic regression analysis (MLRA) using Dataset-C. As a result, we found that the selected parameters in the MLRA model changed appreciably from a set of age, albumin, phosphate, and CVD to a set of BUN, LD, albumin, Ht, and Hb. Therefore, we concluded that a different set of explanatory variables was required for each group (CRP ≥ 40 vs. 20 ≤ CRP < 40 mg/dL) (Table 2, Analysis 4).

We also attempted random resampling from Dataset-A with the same data size (n = 275) (allowing for replacement), and performed MLRA 25 times. We found that the key explanatory parameter (age) was invariably chosen by the stepwise selection of the explanatory parameters (Table 3). However, other parameters, such as BUN and LDH, changed from one dataset to another. We interpreted this finding as follows: (1) The second to lower order parameters were comparable in terms of the power of prediction, which made it difficult to obtain a reproducible set of explanatory parameters; and (2) there exist patients with missing data of some variables who were not used in the regression analysis.

To overcome those problems of the conventional prediction-model-building using MLRA, we devised an alternative method of prediction, the so-called “weighted average of risk scores” (WARS). The WARS is robust in that it allows for missing data in some variables, while also integrating all available risk variables, explained in detail below.

We first determined parameters that may contribute to the prediction by univariate comparison of the two groups (dead/alive), as shown in Fig. 2. 

1) Using ROC analysis for each risk variable, we determined the cutoff value for distinguishing dead cases from alive cases, as shown in Table 1/Fig. 2.

2) Ten risk variables were chosen based on the magnitude of C-statistics (area under the ROC curve), which represents the predictability of mortality risk. Each variable was graded by assigning “weight” (wt) in reference to the boundaries of C-statistics (0.65 and 0.70): wt = 1 (≤0.65), wt = 2 (0.65–0.7), wt = 3 (>0.7), as shown in Table 1.

3) We performed binary transformation of each risk variable (based on the respective cutoff value) for calculating the cumulative score. For example, when the values of the dead group are shifted to a higher side, score 1 is assigned for any value above the cutoff; otherwise, the score is set to 0. When the values of the dead group are shifted to a lower side, any value below the cutoff is assigned a score of 1.

4) The average risk score for each patient was calculated as 

 WARS= (∑_i^np▒〖〖wt〗_i×sx〗_i )/(∑_i^np▒〖wt〗_i ) 

where sxi represents a score (0 or 1) assigned to the risk parameter i, np represents the number of 

risk variables available for a given patient, and wti indicates a weight assigned to sxi (1, 2, or 3).

The utility of WARS for predicting mortality risk was evaluated using univariate logistic regression analysis. Compared to MLRA, WARS attained the same degree of distinction between the two groups (dead vs. alive) in terms of C-statistics (0.837), as shown in Table 4.　

In addition to the comparable prediction performance, the advantages of WARS were (1) its coverage of a larger number of risk variables in combination, and (2) its applicability, even in the presence of missing data. 

We apologize for the lengthy response. We are very thankful to the reviewers for their constructive comments. We had an opportunity to critically re-appraise our methodology and invented a new scheme for risk prediction.

As a result of these changes, please allow us to make drastic changes throughout the manuscript. 

---------------

General feedback on the manuscript is as follows.

The title and abstract were appropriate.

The introduction was reasonable but the authors need to introduce the concept and implications of extremely high serum CRP in clinical settings and frequency in patients. The authors only note CRP levels in healthy subjects and how it can become elevated 10,000-fold in hospital patients.

Our response→ Following the reviewer’s suggestion, we added the background for setting our target to cases with CRP levels ≥40 mg/dL in the 4th paragraph of the Introduction.

The methods section is robust for most part but there is a large contradiction in the section named “Study design and participant selection” with regards the number of patients included in the study (lines 95-100 contradict with lines 103-108). Also, the occurrence of 0.1% (as stated by the authors) suggests there is only 283 adult patients with extremely high CRP out of a total number of 283,000 adult patients over 18 years old. If so, I presume most of the CRP data list (i.e. the remaining 625, 750 patients) was not useful/relevant? This statement needs clarification to explain how often an adult patient (above 18 years old) had a CRP level over 40 mg/dL. Lastly, the authors should refer the reader to the Tables when stating ‘underlying causes of extremely high CRP’ (lines 123-124) were considered; otherwise those lines are somewhat vague.

Our response→ Thank you for pointing out the inconsistency in numerical values. We have corrected the Methods section according to your suggestions in reflection of the increased sample size as follows. 

“This was a retrospective, single center, case-controlled cross-sectional study. We used a CRP data list that included 1,336,403 patients aged over 18 years who visited Saitama Medical Center between 2004 and 2017. The incidence of an extremely high outlier value of CRP level >40 mg/dL was 0.0302% overall, and 401 patients were selected. This rate meets the extreme outlier values that lie statistically outside the 0.5.99.5 percentile range [4]. We excluded 113 records that were metachronous duplicates from the same patient (only one highest value of CRP from each of these patients was considered), 10 patients with cardiopulmonary arrest at arrival, and 6 patients with unknown outcomes. After applying the exclusion criteria, a sample of 275 patients was selected for use as a training dataset (Dataset-A) to build a model for predicting mortality risk.”

The results section was good for the most part, except the statement on lines 204-205 seems confused to me, and it is not clear why potassium levels were omitted (not considered further) in the Prediction Model (lines 208 onwards).

Our response→ As mentioned above, we drastically changed our article, especially the Methods, Results, and Discussion. Therefore, the original lines have gone. Please allow us to omit our response to this comment.

The limitations need to note that the findings only constitute a very small cohort of patients (especially after applying your exclusion criteria) from the pool of typical clinical CRP samples taken in hospital so the impact/relevance is relatively low & limited despite being of medical importance to affected individuals.

Our response→ Thank you for your comment. We have added additional statements in the Limitations section as follows.

“Second, we selected patients with CRP levels over 40 mg/dL (only 0.0302% of our cohort); therefore, the clinical relevance of our findings for the general population is limited.”

The discussion is reasonable throughout but the authors have neglected to discuss some important findings of the study, including those relating to dementia and significant data relating to biochemical measurements such as potassium, albumin and total cholesterol levels.

Our response→ We appreciate the constructive comments. We have changed the statistical logic of predicting mortality from MLRA to the robust risk scoring scheme (WARS). By using the latter, we have incorporated many biochemical risk parameters, including potassium and albumin, which had been suppressed in the initial MLRA because of the presence of closely associated risk parameters. In the revised Discussion, we emphasize the relevance of the inclusion of biochemical risk variables and the need to reflect all of them in predicting mortality.

The conclusion is appropriate for the most part but to avoid overplaying the findings in lines 293-295 the authors need to reiterate that the utility of the proposed prediction model to enable physicians to easily estimate the 72-hour mortality probability and facilitate an easier decision-making process only applies to adult patients falling outside your designated list of exclusion criteria.

Our response→ Thank you for your advice. Given the drastic change in the manuscript, the Conclusion was fully rewritten in consideration of the suggestion as below. 

“We aimed to develop a numerical model for predicting 72-hour mortality in patients with a disputed CRP level of ≥40 mg. Multiple risk variables were identified by univariate analysis, including increased age, IP, BUN, Mg, K, LDH, DD, low Alb and fibrinogen, and presence of CVD. By MLRA, the selected variables in the optimal regression model were limited to age, Alb, IP, and CVD; this was due to similar predictability among other variables and the presence of many missing data in some variables. In contrast, mortality prediction using WARS resulted in a performance comparable to that of MLRA, and exhibited a superior property of covering all relevant risk variables and robustness in use even with missing data. Therefore, we believe that the novel WARS should be used as the method of choice for predicting 72-hour mortality in patients with extremely high CRP levels. WARS has the potential to assist physicians in decision-making, provide therapeutic and management options to patients and their families, and improve the quality of initial medical management in both primary and critical care settings.”

There were some minor typological and spelling errors noted (see file attached).

Our response→ Thank you for your corrections. We have implemented the suggested changes.

Reviewer #2: CRP is a well-established marker of systemic inflammation. Because the levels of CRP rise much more significantly during acute inflammation than the levels of the other acute phase reactants, the CRP test has been used for decades to indicate the presence of systemic inflammation, infection, or sepsis.

The present study is the hospital-based study to investigate the risk factors associated with 72-hour mortality in people with extremely high CRP. The main findings were that the 72-hour mortality rate in patients with extremely high CRP was 14.2% and that the main independent risk factors associated with mortality were age, serum inorganic phosphorus, and blood urea nitrogen. level.

The manuscript is interesting, but suffers with some limitations. The main questions to the authors are:

Our response→ Thank you for critically reviewing our manuscript and providing us with comments for making revisions.

1. It is not clear why the value of CRP over 40 mg/dl was set as cut-off? According to Póvoa the CRP levels in patients with sepsis, severe sepsis and septic shock were 15.2 ± 8.2, 20.3 ± 10.9 and 23.3 ± 8.7 mg/dL. [Póvoa P, et al. C-reactive protein as a marker of infection in critically ill patients. Clin Microbiol Infect. 2005]. 

Moreover, according to the reference 22 (Line 96) high value of CRP is not a good marker of the mortality rate prognosis. The ICU mortality rate of septic patients with CRP < 10, 10-20, 20-30, 30-40 and >40 mg/dL was 20, 34, 30.8, 42.3 and 39.1%, respectively (P = 0.7). So there is no reason to choose the cut-off value of 40 mg/dl.

Our response→ Thank you for the important comments. We have revised the Introduction section as follows.

“In hospital settings, physicians may encounter patients with CRP levels that significantly exceed the normal range [11]. Indeed, several case reports have reported CRP levels >40 mg/dL in adults with pyogenic liver abscess with complicated intestinal tuberculosis [12] and immune-hemolytic anemia [13]. Silvestre et al. observed no significant differences in CRP concentrations at ICU admission between survivors and non-survivors (25.3 ± 13.7 versus 28.2 ± 13.2 mg/dL). Furthermore, the ICU mortality rates of patients with sepsis during an ICU stay with CRP concentrations <10, 10–20, 20–30, 30–40, and >40 mg/dL were 20%, 34%, 30.8%, 42.3%, and 39.1%, respectively. This finding suggests that CRP is a poor marker of prognosis and that CRP levels >40 mg/dL are not associated with increased mortality of patients with sepsis during ICU stay [14]. In contrast, other studies have reported that CRP level >40 mg/dL indicate severe bacterial infection [15,16], can be used as a predictor of renal scarring associated with a first urinary tract infection [17], and indicate acute pyelonephritis in the pediatric field [18]. Moreover, the cutoff value of CRP level >40 mg/dL was used to predict sepsis in emergency departments and demonstrated a sensitivity of 82.3% and specificity of 38.7% [19]. Furthermore, the use of a point-of-care CRP level >40 mg/dL significantly increases the predictive accuracy of treatment failure among patients with exacerbated mild to moderate chronic obstructive pulmonary disease [20, 21].”

2. It would be important to have more baseline clinical data about the patient population (e.g. medication).

Our response→ Thank you for your suggestion, we have added Suppl Table S1 to show the baseline clinical data on medications. Cholinesterase inhibitors showed statistical differences between the two groups (4.5% vs. 0.0%). The other medications were not significant.

3. Small sample size. Authors calculated the required sample size as 23 deaths and 138 survivals to achieve a power of 80% and an alpha error of 5%. The 72-hour mortality in the investigation was 26 and three variables were included into the multivariate logistic regression analysis. It is recommended to use at least 10 events per variable to fit prediction models in clustered data using logistic regression. 

Up to 50 events per variable may be needed when variable selection is performed. [Wynants L, et al. A simulation study of sample size demonstrated the importance of the number of events per variable to develop prediction models in clustered data. J Clin Epidemiol. 2015 Dec;68(12):1406-14.]. 

Age (years), serum inorganic phosphorus (mg/dL) and blood urea nitrogen value (mg/dL) are continuous variables. Into the investigation were included only 26 events. 

So the power of the investigation is not sufficient.

Our response→ As recommended by Reviewer-1, we extended the study period by 3 years and increased the data size to 275 (44 deaths and 233 survivals). We somehow acquired a minimum of 10 events per variable in the regression model built with the entry of four explanatory variables. Although we admit that it was not possible to attain the stricter criteria of Wynants et al., we evaluated the reproducibility of our prediction model using a bootstrap method and demonstrated its reliability at a practical level. 

We believe that it is difficult to attain sufficient reproducibility of variable selection by conventional logistic regression analysis. Therefore, in this revised manuscript, we propose an alternative robust method by using the weighted average of risk scores, which ensures that all relevant risk variables are reflected, even in the presence of missing data. 

In the revised paper, we compare the conventional method with the novel method in terms of their performance in predicting mortality risk. We hope that this method comparison is relevant to the readers of your journal who are interested in developing a generally applicable risk prediction model.

---

## [Editor Report · Decision Letter 1]

18 Jan 2021

Prediction of 72-hour mortality in patients with extremely high serum C-reactive protein levels using a novel weighted average of risk scores

PONE-D-20-07042R1

Dear Dr. Sugawara,

We’re pleased to inform you that your manuscript has been judged scientifically suitable for publication and will be formally accepted for publication once it meets all outstanding technical requirements.

Kind regards,

Pablo Garcia de Frutos

Academic Editor

PLOS ONE
---

## [Editor Report · Acceptance letter]

25 Jan 2021

PONE-D-20-07042R1 

Prediction of 72-hour mortality in patients with extremely high serum C-reactive protein levels using a novel weighted average of risk scores 

Dear Dr. Sugawara:

I'm pleased to inform you that your manuscript has been deemed suitable for publication in PLOS ONE. Congratulations! Your manuscript is now with our production department. 

Kind regards, 

on behalf of

Dr. Pablo Garcia de Frutos 

Academic Editor

PLOS ONE